# Curcumin-Rich Curry Consumption Is Associated with Lower Risk of Cognitive Decline and Incidence of Mild Cognitive Impairment or Dementia: An Asian Population-Based Study

**DOI:** 10.3390/nu17152488

**Published:** 2025-07-30

**Authors:** Yanxia Lu, Tih Shih Lee, Wee Shiong Lim, Philip Yap, Chin Yee Cheong, Iris Rawtaer, Tau Ming Liew, Xinyi Gwee, Qi Gao, Keng Bee Yap, Tze Pin Ng

**Affiliations:** 1Department of Medical Psychology and Ethics, School of Basic Medical Sciences, Cheeloo College of Medicine, Shandong University, Jinan 250012, China; a0068932@u.nus.edu; 2Neuroscience and Behavioral Disorders Program, Duke-NUS Medical School, Singapore 169857, Singapore; tihshih.lee@duke-nus.edu.sg; 3Institute of Geriatrics and Active Aging, Department of Geriatric Medicine, Tan Tock Seng Hospital, Singapore 308433, Singapore; wee_shiong_lim@ttsh.com.sg; 4Lee Kong Chian School of Medicine, Nanyang Technological University, Singapore 308232, Singapore; 5Department of Geriatric Medicine, Khoo Teck Puat Hospital, Singapore 768828, Singapore; yap.philip.lk@ktph.com.sg (P.Y.); cheong.chin.yee@ktph.com.sg (C.Y.C.); 6Department of Psychiatry, Sengkang General Hospital, Singapore 544886, Singapore; iris.rawtaer@singhealth.com.sg; 7Department of Psychiatry, Singapore General Hospital, Singapore 169608, Singapore; liew.tau.ming@singhealth.com.sg; 8Gerontological Research Programme (GRP), Department of Psychological Medicine, Yong Loo Lin School of Medicine, National University of Singapore, Singapore 119074, Singapore; pcmgxy@nus.edu.sg; 9Department of Clinical Epidemiology, Tan Tock Seng Hospital, Singapore 308433, Singapore; aprilgaoqi@gmail.com; 10Department of Medicine, Ng Teng Fong General Hospital, Singapore 609606, Singapore; keng_bee_yap@nuhs.edu.sg; 11Geriatric Education and Research Institute, Singapore 768024, Singapore

**Keywords:** curcumin, curry, neurocognitive disorder, population-based study

## Abstract

Background/Objectives: We studied the possible protective effect of dietary curcumin in curry meals against cognitive decline and mild cognitive impairment (MCI) and dementia in a population-based Singapore Longitudinal Ageing cohort study. Methods: Baseline curry consumption frequency was categorized as five categories ranging from ‘never or rarely’ to ‘daily’. Among 2920 participants (mean age 65.5 ± SD 7.1 years) free of stroke, Parkinson’s disease, or traumatic brain injury at baseline, cognitive decline (MMSE drop ≥2) was assessed at 3–5 years (mean 4.5) follow-up. Occurrence of incident MCI-dementia was assessed at follow-up among 2446 participants without neurocognitive disorder at baseline. Results: A decreasing linear trend was observed between higher levels of curry consumption and cognitive decline (*p* = 0.037). The cumulative incidence of MCI-dementia decreased from 13.1% in those who never or rarely consumed curry to 3.6% in those who consumed curry daily (linear *p* < 0.001). The adjusted OR across levels of curry consumption exhibited a linear trend (*p* = 0.021) from OR = 0.61 (*p* < 0.05) for occasional consumption to OR = 0.21 (*p* < 0.001) for daily consumption. Conclusions: The intake of dietary curcumin through curry shows a dose-dependent reduction in incidence of cognitive decline and MCI-dementia in this Asian population of community-based elders.

## 1. Introduction

Experimental studies have suggested that curcumin, a polyphenol isolated from turmeric roots, exhibits potent antioxidant, anti-inflammatory, antimicrobial, antiviral, anti-neoplastic, and anti-aging activities [1,2,3]. Its therapeutic potential is intensively investigated in over 100 clinical trials for chronic diseases such as neoplasms, diabetes, obesity, and disorders affecting the cardiovascular, respiratory, nervous, and immune systems [1].

Turmeric, from which curcumin is extracted, is widely used as dyes and traditional Indian and Chinese medicinal herbs. As a food additive, it is the predominant source of curcumin in Asian diets. In the form of the Indian spice, turmeric is consumed in curry meals by millions of people in sub-continental India, Southeast Asia, and other areas of the world. Only limited naturalistic studies have evaluated the possible benefits of dietary food source(s) of curcumin on human cognition. We first showed in a population-based observational study of middle aged and elderly individuals living in the community that a cross-sectional association of curry consumption is found to be associated with better cognition, as examined via the Mini-Mental State Examination (MMSE) [4]. Subsequent longitudinal observations in the Singapore Longitudinal Ageing Study (SLAS) follow-up cohort showed that the intake of dietary curcumin is related to the maintenance of higher functions such as attention, working memory, visuospatial constructional ability, language, and executive function over time. The effect sizes ranged from 0.13 and 0.30 in sub-groups of participants [5].

This report presents the results of the longitudinal analysis of the association of curry consumption with the development of cognitive decline and mild cognitive impairment (MCI) and dementia from 4.5 years of follow-up of older adults dwelling in the community in the SLAS prospective cohort study.

## 2. Materials and Methods

### 2.1. Study Design and Participants

We have described the study design and methodology of the SLAS in previous publications [6]. Briefly, the SLAS cohort included Singapore citizens aged 55+ years residing in designated study areas, excluding individuals with severe physical/mental impairments. The inclusion criteria are: (1) age ≥55 years; (2) non-institutionalized residence; (3) independent ambulation; and (4) absence of physically limiting comorbidities. The exclusion criteria are: (1) significant vision/hearing deficits; (2) progressive neurological disorders (e.g., Parkinson’s disease, epilepsy); (3) terminal conditions with <12-month prognosis under palliative care; and (4) recent hospitalization within 6 weeks. The SLAS included two combined population cohorts: 2804 participants recruited in SLAS 1 from Southeast Singapore during September 2003 to December 2004, and 3270 participants collected in SLAS 2 from Southwest Singapore during March 2009 to June 2013. The initial follow-up assessments occurred 3–5 years post-baseline (mean interval: 4.5 years), with SLAS 1 being conducted from March 2005 to September 2007 and SLAS 2 spanning January 2013 to August 2018.

Ethics clearance was granted in 2019 by the National University of Singapore Institutional Review Board (Protocol 04–140), in compliance with the Declaration of Helsinki and Belmont Report regulations Written informed consent was obtained from all study participants prior to their inclusion in the research.

### 2.2. Measurements

Trained nurses gathered a comprehensive range of socio-demographic characteristics, lifestyle factors, behavioral patterns, psychological measures, and medical/health information from participants during recruitment from face-to-face questionnaire interviews during home visits and clinical and physiological measurements performed at a local study site.

#### 2.2.1. Curry Consumption

During initial assessments and subsequent follow-ups, participants reported their curry consumption frequency using standardized categories: never or rarely (<1/year), occasionally (>1/year, <1/month), often (>1/month, <1/week), very often (≥1/week), or daily. An additional question asked whether curry consumption had remained mostly unchanged or had changed during follow-up. Data indicating consistent responses at baseline and follow-up interviews were used to assign participants unequivocally to the highest and the lowest consumption category. Participants who consistently reported ‘occasional’ or ‘often’ consumption frequencies at both baseline and follow-up were also assigned as such to the appropriate category. For inconsistent responses that differed by one response category between baseline and follow-up, we used the higher consumption response to assign to the nearest ‘occasional’ or ‘often’ categories. For more widely inconsistent responses, we used the average response score to assign them to the nearest response category, either ‘occasional’ or ‘often’.

#### 2.2.2. Diagnosis of Neurocognitive Disorder (MCI and Dementia)

To determine cases with MCI or dementia, participants went through a two-staged procedure of cognitive screening and assessment, followed by consensus-based diagnostic evaluations using DSM-5 diagnostic standards performed by a multidisciplinary team comprising three specialists of geriatricians and psychiatrists.

Cognitive assessments were conducted with culturally adapted MMSE instruments that had undergone local validation in English, Chinese, and Malay translations, with the application of appropriate education- and ethnic-stratified cut-offs [7]. Subjective cognitive complaint was determined from self-report (“Do you feel you have more problems with memory than most?”) and informant reports (IQCODE) of memory and cognitive problems [8,9].

Post-screening cognitive assessment was performed using the Clinical Dementia Scale (CDR) and a battery of neurocognitive tests, which included attention (digit span forward and backward) and spatial span forward and backward memory (Rey Auditory Verbal Learning Test immediate and delayed recall, visual reproduction immediate and delayed recall), executive function (symbol digit modality test, design fluency, and Trail Making Test Part B), language (categorical verbal fluency), and visuospatial abilities (block design). Neurocognitive assessments provide essential diagnostic utility for identifying and classifying neurodegenerative disorders across disease progression phases. It takes 1.5 to 2 h to complete the entire battery, which was conducted in the participant’s primary language, with options including English, Mandarin, or other Chinese dialects. Domain scores were finally transformed to Z scores. More information on the neurocognitive tests and corresponding reference standards can be found in a previous publication [10].

MCI diagnosis was based on published criteria [11]: (1) subjective memory and cognitive difficulties, or IQCODE score above 3.3; (2) objective cognitive impairment in ≥1 or more domains: An MMSE global score of 24–27, or ≥2-point decline from baseline; and one or more neurocognitive domain (attention, memory, executive function, language or visuospatial abilities) scores 1 to 2 standard deviations (SDs) less than the age and education-adjusted mean values, or drop from the baseline of 0.5 SD during follow-up assessments; (3) CDR global score equal to or above 0.5 [12]; (4) maintaining functional independence for basic activities of daily living (BADL); and (5) not demented.

Diagnosis of dementia was performed based on these criteria: (1) evidence of objective cognitive deficit (MMSE total score equal to or below 23, or neurocognitive domain score that was 2 SDs less than the age and education stratified mean values); and (2) presence of functional disability (requiring assistance with one or more BADL or having a CDR global score of 1 or higher). Participants failing to meet MCI/dementia thresholds were designated as cognitively normal (CN).

Cognitive decline was defined by a drop in MMSE score at follow-up of 2 or more points from baseline.

#### 2.2.3. Covariates

Measurements of potential confounders included sex, age (years), ethnicity (Chinese versus non-Chinese, which includes Malay, Indian, and Other), and education (none, 1–6 years, above 6 years). Health behavioral factors included smoking (never, past smoker, and current smoker), alcohol (equal to or above once/week), and physical activity score, social activity score, and productive activity score, which were calculated from the number and frequencies of usual participation in 18 different categories of physical, social, and productive activities [6]. Cardio-metabolic factors included central obesity (waist circumference equal to or above 80 cm in women and equal to or above 90 cm in men) [13]. Hypertension was defined by systolic blood pressure ≥ 140 mmHg and/or diastolic blood pressure ≥ 90 mmHg [14], or a self-report history of hypertension diagnosis and treatment, verified by the recorded names of drugs shown on medication packages. Diabetes was defined by fasting blood glucose concentrations of ≥5.6 mmol/L [15] or a self-report history of diabetes diagnosis and treatment. Dyslipidemia was defined as either triglyceride levels of equal to or above 1.7 mmol/L or high-density lipoprotein cholesterol below 1 mmol/L in men and below 1.3 mmol/L in women [16]. Prevalent cardiovascular disease included self-report histories of stroke, coronary disease, myocardial infarct, or heart failure. The assessment of depressive symptoms was conducted using the locally adapted and validated 15-item Geriatric Depression Scale (GDS) [17]. As the strongest genetic risk factor for cognitive function, apolipoprotein E (APOE) genotyping was performed and APOE-ε4 ≥ 1 allele was categorized as the APOE-ε4 allele carrier.

### 2.3. Analysis

In the original cohort of 6074 participants, a total of 5035 participants were free of stroke, Parkinson’s disease, or traumatic brain injury and had available data on curry intake. Longitudinal analyses were performed on data for cognitive decline in 2920 participants with data available for MMSE at follow-up (432 who showed cognitive decline and 2489 who did not show cognitive decline); see flow chart in Figure 1. After excluding 101 participants who had dementia, 626 participants who had MCI, and 149 with unknown neurocognitive status, there were 4159 cognitive normal participants at baseline. Longitudinal data analysis for incident MCI or dementia involved 2446 participants free of MCI and dementia at baseline and provided available data on mild cognitive impairment or dementia status at follow-up.

Statistical comparisons of baseline characteristics were performed using ANOVA tests for continuous measures and χ^2^ tests for categorical variables, contrasting participants who developed cognitive decline or mild cognitive impairment or dementia versus those who did not. The odds ratios (ORs) with their 95% confidence intervals (CIs) of associations of curry intake with cognitive decline, incident mild cognitive impairment, or dementia were examined in binary logistic regression models adjusting for the baseline potential confounding factors, as described above. Data analyses were conducted using IBM-SPSS version 25. All hypothesis testing employed two-tailed analyses, with an α-level of 0.05 considered significant.

## 3. Results

In this study, the 2920 study participants had a mean (±SD) age of 65.5 (±7.1) years, and 65.2% were females. There were more participants who reported higher levels of curry intake (Table 1). Baseline characteristics of SLAS study participants (N = 2920) by curry consumption levels at baseline were likely to be younger, men, non-Chinese (Malay or Indian) ethnicity, better educated, have higher levels of social and productive (mentally stimulating) activity, have more central obesity, have diabetes, and have lower GDS depression scores (Table 1). Greater frequency of curry intake was significantly associated with higher mean levels of MMSE global cognition.

### 3.1. Cognitive Decline

Curry consumption was inversely associated with cognitive decline (Table 2 and Table 3). Increasing levels of curry consumption were linked to decreased likelihood of cognitive decline, from 18.7% in those who never or rarely consumed curry to 12.8% who consumed curry daily, *p* for linear trend = 0.037. Controlling for the influence of baseline confounders, the adjusted OR of association showed a similar linear trend of association from 0.73 to 0.66 (*p* = 0.15). (Table 3 and Figure 2) The lowered OR of association with cognitive decline was significant for consuming curry often (OR = 0.68, 95%CI = 0.48–0.95) compared with never or rarely consuming curry.

### 3.2. Incident MCI-Dementia

Among the 2446 participants who were cognitively normal at baseline, an increasing level of curry consumption was inversely associated with the development of incident MCI-dementia, *p* for linear trend < 0.001. (Table 2) The cumulative incidence of MCI-dementia decreased from 13.1% in those who never or rarely consumed curry to 3.6% in those who daily consumed curry (Table 4 and Figure 2). Controlling for the influence of baseline confounders, the adjusted OR across increasing levels of curry consumption exhibited a linear trend (*p* = 0.021), OR = 0.61 (*p* < 0.05) for occasional consumption compared with OR = 0.21 (*p* < 0.001) for daily consumption.

## 4. Discussion

This study indicates that dietary curcumin through curry consumption is associated with a dose-dependent decrease in incidence of cognitive decline and MCI-dementia in this Asian population of community-dwelling older adults. Our results are consistent with the findings from animal studies [18,19]. Results from animal studies consistently demonstrate the positive effects of curcumin and its analogs on the maintenance of cognitive function. In senescence-accelerated mouse (SAM)-prone 8 (SAMP8) mice, intragastric administration of curcumin (20 or 50 mg/kg) for a duration of 25 days is associated with a dose-dependent effect on decreased latency in the Morris water maze (MWM), whereby mice given the highest dose had the same latency as that reported with the control group SAMR1 mice [18]. Qi Chen and colleagues [19] investigated the cognitive effects of J147, a structural analog of curcumin in young Sprague Dawley (SD) rats and C57 BL/6 mice. They found that in young SD rats, J147 (at a dose of 10 micromolar for a duration of 2 wks) promoted long-term potentiation (LTP) induction in the hippocampus and improved the cognitive performance in a novel object recognition test. J147-supplemented C57 BL/6 mice demonstrated an improvement in spatial memory in novel object location (NOL), short-term memory on the Y-maze, and exhibited fewer errors than control mice in the Barnes maze’s retention test portion.

However, randomized trials of curcumin intake for cognition enhancement in non-demented elderly has generated mixed results. In a randomized, double-blind, placebo-controlled trial involving 60 cognitively intact elderly participants aged above 60 years, a lipid curcumin formulation (400 mg as Longvida^®^) improved working memory after both acute (1 and 3 h after a single dose) and chronic (4 weeks) administration. Participants receiving curcumin performed significantly better on the digit vigilance task after acute treatment. Participants receiving both the acute and chronic curcumin treatment performed better than the placebo on the serial-3 subtraction task [20].

Rainey-Smith et al. [21] assessed the efficacy of curcumin (1500 mg/d BiocurcumaxTM) in non-demented elderly in a 48-week randomized, placebo-controlled, double-blind study. There was a significant difference in cognition measures observed at 24 weeks, where the individuals receiving curcumin performed significantly better than the placebo group on the Montreal Cognitive Assessment (MoCA). However, no significant between-group difference was found at 48 weeks when the intervention was completed. These findings suggest curcumin’s potential as a therapeutic intervention for cognitive decline, although the findings remain inconclusive at this time.

Studies have demonstrated the Alzheimer’s disease (AD)-modifying effects of curcumin [2]. In mice with AD (APP/PS1 double knockout model), a 6-month curcumin diet was associated with better working and long-term memory performance on the Morris water maze in a dose-dependent manner (low dose: 160 ppm; high dose: 1000 ppm). The aggregation of amyloid beta 42 (Aβ 42) in mouse brain also decreased, and the clearance rate of dissolved aggregates was better. The number of autophagosomes [22,23] in the CA1 region of the curcumin group significantly increased, while the expression of Beclin 1 increased and the PI3 K/Akt/mTOR signaling pathway was downregulated [24]. Curcumin could protect neuron-like PC12 rat cells and umbilical endothelial cells from Aβ damage by inhibiting oxidative damage and excessive hyperphosphorylation of tau protein [25], promote the uptake of Aβ by macrophages of AD patients [26], and dose-dependently reduce the formation and extension of fibroblasts, while disrupting the stability of pre-formed Aβ fibrils [27,28,29]. Furthermore, curcumin can reduce Aβ-induced radical oxygen species [30] and suppresses amyloid precursor protein (APP) cleavage [31]. It is worth noting that curcumin restores synaptic plasticity via the enhancement of long-term potentiation in rat hippocampal slices treated with Aβ oligomers [32]. Since curcumin clears Aβ and tau and promotes autophagosomes in the brain, this may also be a target for future clinical research.

Regarding the different forms of dementia, existing studies have investigated the effects of curcumin on overall dementia and its major form Alzheimer’s disease, with reports of inconsistent findings [33,34]. The first study that reported the beneficial role of curcumin in cognitive function originates from an epidemiological study conducted in India. In this study, Ganguli and collaborators showed that the Indian population, who consumes a diet rich in curry, have a lower prevalence of AD than the USA population [35]. We also found in previous study that more frequency curry consumption in older healthy individuals is associated with better cognitive function [4]. Curry can delay cognitive decline in older individuals with mild cognitive impairment and AD [36]. Curry attenuates Aβ cytotoxicity and suppresses pathological tau phosphorylation [25], enhances macrophage-mediated Aβ clearance in AD patients [26], and dose-dependently inhibits both nucleation and elongation of Aβ fibrils [27,28,29]. A recent meta-analysis revealed that curry consumption promotes cognitive function in older adults and Asian participants in comparison with younger adults and Western ones [33,37]. It warrants investigation of the effects of curry consumption on other forms of dementia, such as vascular dementia and frontotemporal dementia.

### Strengths and Limitations

This research captures real-world cognitive outcomes in a sizable cohort of non-institutionalized seniors monitored over a median 4.5-year period. The follow-up duration is relatively short. While this is appropriate for estimating adequate numbers of persons progressing from normal cognition to incident MCI, the number of incident cases of dementia was small and should be more accomplished with longer periods of follow-up. Due to the difficulty in monitoring how much the participants consume, we only collected information about the frequency of consumption. Although we excluded participants with stroke, Parkinson’s disease, or traumatic brain injury at baseline and adjusted for major diseases and medication taking in other participants, we could not fully exclude the possibility of the influence of other medications or neurotrophic substances taken by the participants. The observed estimates of risk associated with exposure to curry consumption are likely to be underestimated due to survival bias. This is because the study selectively followed up participants with relatively healthy risk factor profiles and favorable cognitive statuses. A notable strength is that incident cases of MCI and dementia were determined using rigorous clinical assessment and consensus panel diagnosis. One should cautiously generalize the findings to other populations, considering that the study was conducted in a population of Asian ethnicity with higher levels of curry consumption. Nevertheless, it is interesting to surmise that populations with relatively lower prevalence and modest levels of curry consumption could experience similar cognitive benefits, given that the significant effects were observed at moderate levels of curry consumption. A non-linear dose-dependent effect of curcumin exposure is plausible and interesting because laboratory evidence has documented a pronounced hermetic effect of curcumin on diverse biological processes, exhibiting biphasic dose responses, with low doses having stronger effects than high doses for some effects [38]. This has also been observed with higher cognitive performance at relatively lower levels of curry consumption in a previous study [5].

## 5. Conclusions

Our results support the cognitive benefits of curcumin from natural dietary food sources in humans. They suggest that increased intake of curcumin in a turmeric-rich diet could enhance population health by reducing the rate of cognitive decline and risk of developing MCI or dementia. Further studies should be conducted to replicate these findings in other similarly exposed populations.

## Figures and Tables

**Figure 1 nutrients-17-02488-f001:**
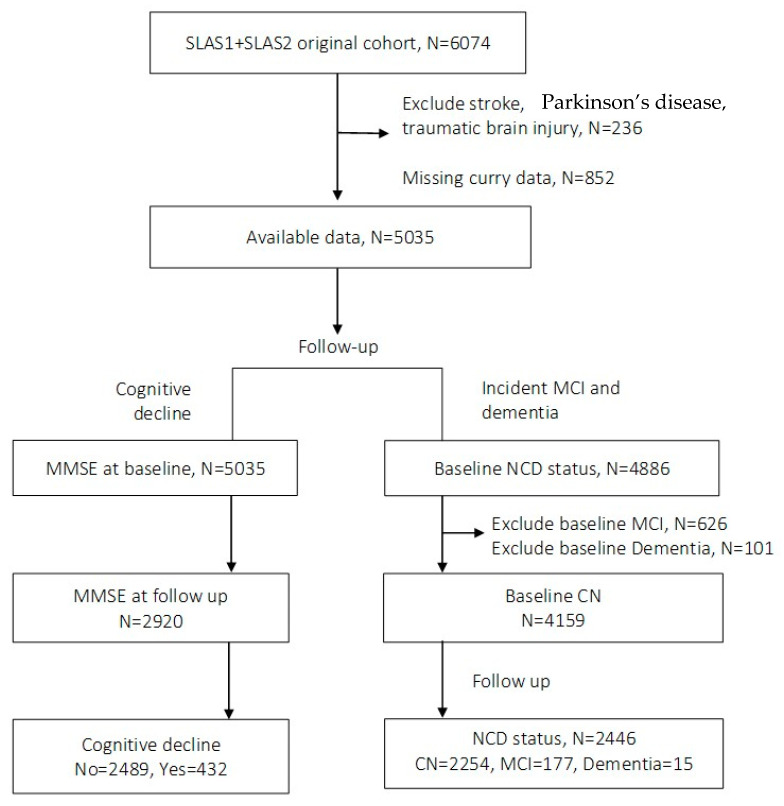
Flow chart of analytical data.

**Figure 2 nutrients-17-02488-f002:**
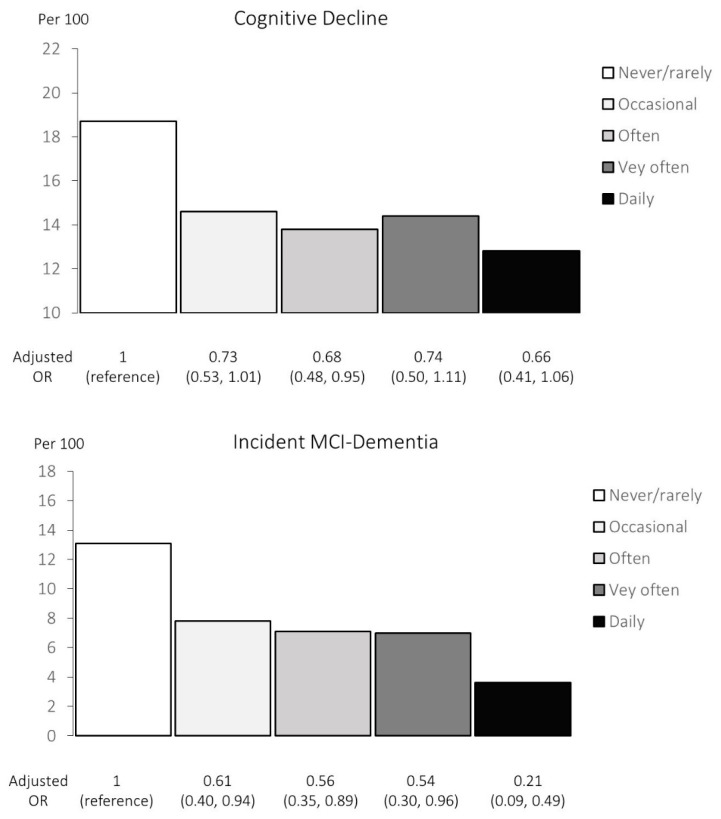
Rates of cognitive decline and incident MCI-dementia by frequency of curry consumption.

**Table 1 nutrients-17-02488-t001:** Baseline characteristics of SLAS study participants (N = 2920) by curry consumption levels at baseline.

	Never or Rarely	Occasional	Often	Very Often	Daily	
	(Never or < Once/Year)	(>Once/Year, <Once/Month)	(>Once/Month, <Once/Week)	(>Once/Week, Not Daily)	(≥Once Daily)	*p*
N of participants	417	1030	805	395	273	
Sex: women	71.5 (298)	70.6 (727)	63.2 (509)	58.0 (229)	52.0 (142)	<0.001
Age, years	66.4 ± 7.3	65.7 ± 7.0	65.6 ± 7.2	64.9 ± 7.1	64.4 ± 6.7	0.003
Ethnicity: Chinese	98.6 (411)	97.4 (1003)	93.4 (752)	89.9 (355)	75.5 (206)	
Malay, Indian, and Other	1.4 (6)	2.6 (27)	6.6 (53)	10.1 (40)	24.5 (67)	<0.001
Education: 0–6 years	63.3 (264)	53.2 (548)	48.9 (394)	44.6 (176)	42.5 (116)	<0.001
Smoking: past smoker	8.4 (35)	6.7 (69)	9.7 (78)	11.4 (45)	8.4 (23)	
Current smoker	5.0 (21)	6.3 (65)	7.8 (63)	6.3 (25)	9.5 (26)	0.005
Alcohol ≥ once/week	5.3 (22)	4.1 (42)	2.2 (18)	5.3 (21)	8.1 (22)	0.143
Physical activity score	2.45 ± 1.91	2.42 ± 1.82	2.30 ± 1.70	2.39 ± 1.84	2.43 ± 1.88	0.594
Social activity score	2.87 ± 2.30	3.32 ± 2.48	3.28 ± 2.54	3.30 ± 2.44	3.35 ± 2.60	0.023
Productive activity score	3.96 ± 1.76	4.02 ± 1.73	4.05 ± 1.85	4.18 ± 1.82	4.15 ± 1.92	0.375
APOE-ε 4 ≥ 1 allele	16.3 (68)	17.6 (181)	15.7 (126)	15.0 (65)	18.3 (50)	0.901
Central obesity	43.2 (180)	48.7 (502)	52.7 (424)	50.1 (198)	52.7 (144)	0.009
Hypertension	59.2 (247)	59.2 (610)	62.2 (501)	56.7 (224)	54.6 (149)	0.256
Diabetes or FBG > 5.6 mmol/L	17.3 (72)	17.8 (183)	22.5 (181)	19.0 (75)	22.7 (62)	0.032
High triglyceride > 2.2 mmol/L	18.0 (75)	27.6 (284)	33.4 (269)	21.0 (83)	26.0 (71)	0.167
Low HDL-cholesterol (<1.0 mmol/L)	18.7 (78)	27.2 (280)	34.7 (279)	19.2 (76)	29.3 (80)	0.068
Cardiac diseases	8.2 (34)	6.3 (65)	8.2 (66)	5.6 (22)	8.8 (24)	0.819
GDS depression score	1.58 ± 2.30	1.01 ± 1.94	0.91 ± 1.74	1.27 ± 2.30	1.33 ± 2.42	<0.001
GDS ≥ 5	9.8 (41)	5.8 (60)	3.6 (29)	7.6 (30)	7.7 (21)	<0.001
MMSE	27.3 ± 3.0	27.9 ± 2.5	28.2 ± 2.2	28.1 ± 2.3	27.8 ± 2.8	<0.001

**Table 2 nutrients-17-02488-t002:** Baseline levels of curry consumption and covariates by cognitive decline status (N = 2920) and incident mild cognitive impairment or dementia status (N = 2446).

		Cognitive Decline			Incident Mild Cognitive Impairment or Dementia		
		No	Yes	*p*	No	Yes	*p*
No. of participants		N = 2489	N = 432		N = 2254	N = 192	
Never or rarely	never or <once/year	13.6 (339)	18.1 (78)	0.037	12.9 (291)	22.9 (44)	<0.001
Occasionally	>once/year, <once/month	35.4 (880)	34.8 (78)		35.5 (801)	35.4 (68)	
Often	>once/month, <once/week	27.9 (694)	25.8 (150)		28.4 (641)	25.5 (49)	
Very often	>once/week, not daily	13.6 (338)	13.2 (57)		13.6 (307)	12.0 (23)	
Daily	≥once daily	9.6 (238)	8.1 (35)		9.5 (214)	4.2 (8)	
Sex	Female	64.3 (1601)	70.5 (304)	0.012	63.8 (1439)	73.4 (141)	0.008
Age, years	Mean ± SD	65.2 ± 6.9	67.4 ± 7.8	<0.001	64.6 ± 6.6	68.8 ± 7.9	<0.001
Non-Chinese ethnicity	Malay, Indian, and Other	6.3 (156)	8.6 (37)	<0.001	5.2 (118)	10.9 (21)	<0.001
Education	0–6 years	49.4 (1229)	62.4 (269)	<0.001	43.5 (981)	72.4 (139)	<0.001
Smoking	Past smoker	9.0 (224)	6.0 (26)	0.056	8.6 (193)	6.8 (13)	0.657
	Current smoker	7.0 (175)	5.8 (25)		7.1 (159)	7.8 (15)	
Alcohol	≥once/week	4.3 (106)	4.4 (19)	0.887	4.5 (101)	4.2 (8)	0.839
Physical activity score	Mean ± SD	2.41 ± 1.81	2.30 ± 1.83	0.258	2.49 ± 1.82	2.10 ± 1.75	0.004
Social activity score	Mean ± SD	3.25 ± 2.47	3.19 ± 2.51	0.653	3.36 ± 2.55	3.10 ± 2.06	0.174
Productive activity score	Mean ± SD	4.09 ± 1.80	3.87 ± 1.80	0.020	4.16 ± 1.80	3.94 ± 1.78	0.097
APOE-ε 4 ≥ 1 allele		16.6 (413)	17.9 (77)	0.514	16.1 (362)	19.3 (37)	0.248
Central obesity		49.0 (1219)	53.1 (229)	0.111	48.8 (1101)	56.8 (109)	0.035
Hypertension		58.3 (1452)	64.7 (279)	0.013	58.4 (1317)	62.5 (120)	0.271
Diabetes or FBG	>5.6 mmol/L	19.7 (490)	19.3 (83)	0.836	17.7 (400)	25.5 (49)	0.010
High triglyceride	>2.2 mmol/L	27.5 (685)	22.5 (97)	0.029	27.9 (628)	21.4 (41)	0.052
Low HDL-cholesterol	(<1.0 mmol/L)	27.6 (686)	24.8 (107)	0.239	28.0 (631)	22.9 (44)	0.131
Cardiac diseases		6.7 (168)	10.0 (43)	0.020	6.7 (151)	6.3 (12)	0.811
GDS depression score	Mean ± SD	1.14 ± 2.05	1.24 ± 2.07	0.326	0.91 ± 1.67	1.47 ± 2.36	<0.001
GDS ≥ 5		6.1 (153)	6.5 (28)	0.781	4.2 (95)	9.9 (19)	<0.001
MMSE	Mean ± SD	27.8 ± 2.55	28.2 ± 2.35	0.009	28.7 ± 1.62	27.2 ± 2.43	<0.001

**Table 3 nutrients-17-02488-t003:** Association of curry consumption with cognitive decline (N = 2920).

		Exposed	Cognitive Decline		Unadjusted				Adjusted †			
		N	Yes	Per 100	OR	95%	CI	*p*	OR	95%	CI	*p*
Never or rarely	never or <once/year	417	78	18.7	1				1			
Occasionally	>once/year, once/month	1030	150	14.6	0.74	0.55	1.00		0.73	0.53	1.01	
Often	>once/month, <once/week)	805	111	13.8	0.69	0.51	0.96	*	0.68	0.48	0.95	*
Very often	>once/week, not daily)	395	57	14.4	0.73	0.50	1.06		0.74	0.50	1.11	
Daily	≥once daily	273	35	12.8	0.64	0.41	0.98	*	0.66	0.41	1.06	
Linear trend, *p*								0.037				0.150

† Adjusted for age, sex, ethnicity, education (≤6 years of schooling), smoking, physical activity score, social activity score, productive activity score, apolipoprotein E-ε 4 allele, central obesity, high fasting blood glucose or diabetes, hypertension, low high-density lipoprotein cholesterol level, high triglycerides, cardiac disease, GDS score, and baseline MMSE. * *p* < 0.05.

**Table 4 nutrients-17-02488-t004:** Associations of curry consumption with incident mild cognitive impairment or dementia among cognitive normal participants (N = 2446).

		Exposed	Mild Cognitive Impairment or Dementia		Unadjusted				Adjusted †				
		N	MCI + Dementia	Per 100	OR	95%	CI	*p*	OR	95%	CI	*p*	
Never or rarely	never or <once/year	335	42 + 2	13.1	1				1				
Occasionally	>once/year, <once/month	869	61 + 7	7.8	0.56	0.38	0.84	**	0.61	0.40	0.94	*	
Often	>once/month, <once/week)	690	46 + 3	7.1	0.51	0.33	0.78	**	0.56	0.35	0.89	*	
Very often	>once/week, not daily)	330	20 + 3	7.0	0.49	0.29	0.84	**	0.54	0.30	0.96	*	
Daily	≥once daily	222	8	3.6		0.25	0.11	0.54	***	0.21	0.09	0.49	***
Linear trend, *p*									<0.001				0.021

† Adjusted for age, sex, ethnicity, education (≤6 years of schooling), smoking, physical activity score, social activity score, productive activity score, apolipoprotein E-ε 4 allele, central obesity, high fasting blood glucose or diabetes, hypertension, low high-density lipoprotein cholesterol level, high triglycerides, cardiac disease, and GDS score. * *p* < 0.05; ** *p* < 0.01; *** *p* < 0.001.

## Data Availability

The datasets used and/or analyzed during the current study are available from the corresponding author on reasonable request. The data are not publicly available due to privacy reasons.

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
