# Peer review of "Curcumin-Rich Curry Consumption Is Associated with Lower Risk of Cognitive Decline and Incidence of Mild Cognitive Impairment or Dementia: An Asian Population-Based Study"

_nutrients, 2025, doi:10.3390/nu17152488_

Round 1
Reviewer 1 Report
Comments and Suggestions for Authors
It is necessary to make the abstract more readable and concise. The introduction is generalist and could be reduced. It is not clear how you explain that the cognitive improvement caused by curry consumption is dose-indipendent. Moreover, can you clarify if this consumption was twice a day or three times a day or not. Then, was it in tablets or in powder or not?
In the end, it is mandatory that you add informations about the medications taken by these patients. Do you know if none of them were taking neurotrophic substances?
Do you have neuroimaging of these patient samples?
Do you think that curry is more effective in preventing some forms of dementia than others?
Can you really re-write the discussion?
Author Response
Reviewer 1
1. It is necessary to make the abstract more readable and concise. The introduction is generalist and could be reduced. It is not clear how you explain that the cognitive improvement caused by curry consumption is dose-independent. Moreover, can you clarify if this consumption was twice a day or three times a day or not. Then, was it in tablets or in powder or not?
The abstract has been revised to make it more readable and concise. The introduction has been reduced. Because we found that higher levels of curry consumption are associated with a decreasing linear trend of cognitive decline and cumulated incidence of MCI-dementia, we concluded that cognitive improvement caused by curry consumption is dose-dependent. Both consumption of twice a day and three times a day are categorized as daily (≥ once daily) curry consumption in this study. As we stated in the text, the predominant source of curcumin is as a food additive in Asian diets, which is natural dietary food sources.
2. In the end, it is mandatory that you add information about the medications taken by these patients. Do you know if none of them were taking neurotrophic substances?
We have added in Discussion that “Although we excluded participants with stroke, Parkinson disease, or traumatic brain injury at baseline and adjusted for major diseases and meditation taking in other subjects, we could not fully exclude the possibility of the influence of other medications or neurotrophic substances taken by the subjects.”
3. Do you have neuroimaging of these patient samples?
We do not have neuroimaging of these participants, which are not highly relevant data with the study aim of this study.
4. Do you think that curry is more effective in preventing some forms of dementia than others?
Currently there is no relevant literature observed.
5. Can you really re-write the discussion?
We have re-written the Discussion.
Reviewer 2 Report
Comments and Suggestions for Authors
In the current article the authors investigated the potential protective effect of dietary curcumin in curry meals against the risk of cognitive decline and incident neurocognitive disorder (mild cognitive impairment and dementia) in the population-based Singapore Longitudinal Ageing cohort study. They found that dietary curcumin through curry consumption is associated with a dose-dependent decrease in incidence of cognitive decline and MCI-dementia in this Asian population of community-dwelling older adults.
Some suggestions:
1. You forgot to add MCI, AD at abbreviations. Please complete. Also please check if all the abbreviations from the article are explained.
- Point 2.1. Study Design and Participants:
-Please define the exclusion/inclusion criteria in detail.- Please add in which year was the ethical approval obtained? You wrote only “Protocol Code: 04–140”-Since the data collection ended in 2018 why are you publishing the data only in 2025? Please clarify.
3. In my opinion, it is also necessary to monitor how much they consume and in what form (food or supplement). It is not enough to monitor how often they consume.
4.What do you mean by:- lines 112-113:“An additional question asked whether curry consumption had remained mostly unchanged or have changed.” -lines 119-120 “For more widely inconsistent responses”Please clarify.
5.Lines 138-139, you wrote “Details of the neurocognitive tests and their normative values have been described in a previous publication [35, 36].” To make the article easier to read, please provide some details.
6. Please explain what is meant by “physical activity score”, “social activity score”, “productive activity score” and “APOE-e4”.
7.Lines 238-239: what amount of J147, a structural analog of curcumin was administrated to rats/mice and how long? Please add.
- For all the studies presented at discussions (on mice/humans) add please the amount of administrated curcumin and the duration of administration.
Author Response
Reviewer 2
In the current article the authors investigated the potential protective effect of dietary curcumin in curry meals against the risk of cognitive decline and incident neurocognitive disorder (mild cognitive impairment and dementia) in the population-based Singapore Longitudinal Ageing cohort study. They found that dietary curcumin through curry consumption is associated with a dose-dependent decrease in incidence of cognitive decline and MCI-dementia in this Asian population of community-dwelling older adults.
Some suggestions:
1. You forgot to add MCI, AD at abbreviations. Please complete. Also please check if all the abbreviations from the article are explained.
We have added MCI and AD abbreviations. We have checked the text to ensure that all abbreviations from the article are explained.
2. Point 2.1. Study Design and Participants:
-Please define the exclusion/inclusion criteria in detail.- Please add in which year was the ethical approval obtained? You wrote only “Protocol Code: 04–140”-Since the data collection ended in 2018 why are you publishing the data only in 2025? Please clarify.
We have added the detailed exclusion/inclusion criteria to Study Design and Participants. The ethical re-approval for the analysis in this study was obtained in 2019. This study (paper) was conducted after the full data had been collected.
3. In my opinion, it is also necessary to monitor how much they consume and in what form (food or supplement). It is not enough to monitor how often they consume.
The predominant source of curcumin is as a food additive in Asian diets, which is natural dietary food sources. We have added in Discussion that not monitoring how much the participants consume is a limitation of this study.
4. What do you mean by:- lines 112-113:“An additional question asked whether curry consumption had remained mostly unchanged or have changed.” -lines 119-120 “For more widely inconsistent responses”Please clarify.
These two places describe whether there is change in curry consumption in follow up in comparison to baseline, and how they are categorized.
5. Lines 138-139, you wrote “Details of the neurocognitive tests and their normative values have been described in a previous publication [35, 36].” To make the article easier to read, please provide some details.
We have added details about neurocognitive tests in the manuscript.
6. Please explain what is meant by “physical activity score”, “social activity score”, “productive activity score” and “APOE-e4”.
We have added in covariates that “physical activity score, social activity score and productive activity score which were calculated from the number and frequencies of usual participation in 18 different categories of physical, social, and productive activities.” ”As the strongest genetic risk factor for cognitive function, Apolipoprotein E (APOE) genotyping was performed and APOE-ε4 ≥1 allele was categorized as APOE-ε4 allele carrier.”
7. Lines 238-239: what amount of J147, a structural analog of curcumin was administrated to rats/mice and how long? Please add.
We have added in Discussion the amount of J147 and how long of the administration.
8. For all the studies presented at discussions (on mice/humans) add please the amount of administrated curcumin and the duration of administration.
We have added in Discussion the amount of administrated curcumin and the duration of administration to all the studies.
Round 2
Reviewer 1 Report
Comments and Suggestions for Authors
The English MUST be improved: i.e. lines 67-73-77-158-292.
The question n°4 is not banal. Do you think that curry is more effective in preventing some forms of dementia than others? Your answer: currently there is no relevant literature observed is NOT acceptable.
Comments on the Quality of English LanguageThe English must be improved.
Author Response
- The English MUST be improved: i.e. lines 67-73-77-158-292.
The typos in lines 67-73-77-158-292 have been corrected. The English throughout the manuscript has been checked and improved as possible.
- The question n°4 is not banal. Do you think that curry is more effective in preventing some forms of dementia than others? Your answer: currently there is no relevant literature observed is NOT acceptable.
We have gone through most recent review articles [1-3], and added a relevant paragraph in Discussion.
[1] Wang W, Zhao R, Liu B, Li K. The effect of curcumin supplementation on cognitive function: an updated systematic review and meta-analysis. Front Nutr. (2025)12:1549509. doi: 10.3389/fnut.2025.1549509
[2]. Tsai IC, Hsu CW, Chang CH, Tseng PT, Chang KV. The effect of curcumin differs on individual cognitive domains across different patient populations: a systematic review and Meta-analysis. Pharmaceuticals. (2021) 14:1235. doi: 10.3390/ph14121235
[3]. Zhu LN, Mei X, Zhang ZG, Xie YP, Lang F. Curcumin intervention for cognitive function in different types of people: a systematic review and meta-analysis. Phytother Res. (2019) 33:524–33. doi: 10.1002/ptr.6257